# Multi-Objective Optimisation in Multi-QoS Routing Strategy for Software-Defined Satellite Network

**DOI:** 10.3390/s21196356

**Published:** 2021-09-23

**Authors:** Yang Wu, Guyu Hu, Fenglin Jin, Siqi Tang

**Affiliations:** College of Command and Control Engineering, Army Engineering University of PLA, Nanjing 210007, China; 13218082261@163.com (Y.W.); fljin@sina.com (F.J.); tangsiqi3036@163.com (S.T.)

**Keywords:** software-defined satellite network, quality of service routing strategy, interval type-2 fuzzy set, Network Simulator

## Abstract

The satellite network plays an increasingly important role in the global communication. With the development of communication technology, quality of service requirements have become more and more complex and diverse and the quality of service routing strategy of software-defined satellite network has become a more and more hot and difficult issue. In this paper, an interval-type-2 fuzzy set routing algorithm is proposed. Firstly, the multi- quality of service routing problem in software-defined satellite network is modeled. Then, the interval-type-2 fuzzy set routing algorithm is proposed to make fuzzy decisions. A series of experiments conducted in Network Simulator (Version 2.35) have proved that the proposed interval type-2 fuzzy set routing algorithm can reduce average delay, increase total throughput and reduce packet drop rate.

## 1. Introduction

With the development of communication technology, satellite network has been becoming more and more important in communication, which has attracted the wide attention of academia and industry [1]. Satellite network has an indispensable role in supporting global coverage. It not only extend the ground network, but also provide reliable data transmission and normal communication in natural disasters, emergency rescue, geological survey and other applications [2,3]. Satellite network is made up of satellites and star links. Satellites are divided into three types, named GEO (Geostationary Earth Orbit, 36,000 km) satellite, MEO (Medium Earth Orbit, 5000–10,000 km) satellite and LEO (Low Earth Orbit, 500–1500 km) satellite. Compared with GEO and MEO satellites, LEO satellites have the advantages of low delay, low energy consumption and low signal attenuation, which is good for real-time communication and reducing the power of the mobile terminals [4,5]. However, with the popularization of LEO satellites, the number of users and the amount of data increase rapidly. There are many connections between LEO satellites and mobile terminals. However, the processing and storage capabilities of LEO satellites are unable to meet the needs of processing and storing these connections. In order to manage conveniently, researchers proposed an architecture called SDSN(software-defined satellite network) [6]. In this paper, we proposed a Multi-objective Optimisation in multi-QoS (quality of service) routing strategy for SDSN.

The rapid development of network has made the demand for satellite networks higher, including the more data and the more kinds of services. An efficient and intelligent routing strategy is needed to meet these multi-QoS requirements. These multi-QoS requirements refer to delay, throughput, and packet drop rate. The routing issues of SDSN is to how to desigh the routing strategy to meet these multi-QoS requirements. In order to consider the multi-QoS requirements, the satellite network routing model was established and the multi-objective optimization was carried out.

Based on the SDSN, this paper establishes a multi-QoS routing model and proposes an efficient multi-QoS routing strategy. We summarize our main results as follows:We modeled the multi-QoS routing problem in the architecture of SDSN. In this paper, virtual topology strategy is adopted to shield the dynamics of satellite motion. The routing strategy is designed for each time slot. It is assumed that there are four neighbors of LEO satellite that does not pass through the polar region and there are four ISLs (Inter-Satellite Links) in total. There are four queues to store packets of four different types of applications in a LEO satellite, including delay-sensitive applications, video applications, packet-drop-sensitive applications and applications without special QoS requirements.Based on the multi-QoS routing model, we propose the interval Type-2 fuzzy set routing algorithm. In the control plane, GEO satellites send control information to update routing table of LEO satellites. In the data plane, when the sending node sends a packet, the packet is put into the queue of the corresponding sending buffer according to its service type. Interval Type-2 fuzzy set is used to make fuzzy decisions when selecting the next hop.Network simulation is carried out on Network Simulator (Version 2.35). By simulating Iridium system, compared with ELB routing algorithm and centralized DSP(Dijkstra’s Shortest Path) routing algorithm, the proposed interval Type-2 fuzzy set routing algorithm can reduce average delay, increase total throughput and reduce packet drop rate.

The paper is organized as follows. Section 2 presents the related works regarding the routing algorithm in satellite networks. Section 3 introduces fuzzy sets theory and interval type-2 fuzzy sets and Interval type-2 fuzzy set has been used in proposed multi-QoS routing strategy. Section 4 introduces multi-QoS routing model. Section 5 describes our interval type-2 fuzzy set routing algorithm in detail. Section 6 gives the simulation results and analysis. Finally, Section 7 gives the conclusion.

## 2. Related Works

Routing strategy has become a hot and difficult issue in satellite network. To simplify, we assume that the topology of the terrestrial network is fixed. The difference between satellite network and terrestrial network is the dynamic network topology. The satellite network topology of Non-Geostationary Earth Orbit satellite constantly change because of its high-speed motion. In particular, the topology of LEO satellite network changes regularly with time. If we study the satellite network routing strategy, we need to find out how to shield the dynamic topology of LEO satellite network. In order to shield the dynamic topology of LEO satellite network, there are three solutions, including virtual topology strategy [7,8,9], virtual node strategy [10,11,12], and the topology dependent mechanism [13].

Virtual topology strategy divides the running period of the constellation into slots and the satellite network topology is unchanged in each slot. There are routing tables calculated in advance and stored in the satellites. When sending data, LEO satellite selects the proper routing table based on the current slot. The method needs to occupy huge storage spaces. Rajagopal et al. [14] studied and designed a new hybrid distributed routing model based on extreme learning machine and multi-task beetle antenna search algorithm. This model is superior to other routing models in average delay and packet drop rate. Zhou et al. [15] proposed a routing strategy based on membership degree, established a routing model based on uncertain link information and used grey wolf optimization algorithm to solve the problem. The algorithm is able to achieve efficient and secure routing in complex environments. [16,17] deduced a strategy with the theory and time is divided into slots. In the process, the dynamic topology of satellite network is divided into static network topologies according to the time slots on the basis of the connection of LEO satellites with their neighbors, then the time slot length and quantity of static network topologies can be calculated. Ref. [18] adopted virtual topology strategy to shield the dynamic characteristics of satellite network, so as to transform dynamic routing into static routing and minimize the complexity of routing protocol.

Virtual node strategy refers to the projection point of each satellite in the initial state as a virtual node, each of which has an independent identity. The satellite will get its identity of nearest virtual node. When satellite changes its position, its identity changes to shield the dynamic topology of LEO satellite network. The method demands the state of ISL keeps steady and is applied to the iridium constellation. Ref. [19] further studied virtual node strategy. When a ground terminal is covered by many LEO satellites, a MSVN (multi-state virtual topology) [19] is proposed on the basis of the fixed footprint model of earth. In order to theoretically analyze and deduce the network topology by using the virtual node strategy, Ref. [12] adopts the formal method to analyze attributes, and optimize the virtual node model.

The topology dependent mechanism refers to the whole LEO satellite constellation is regarded as a two-dimensional plane. The relative logical positions of all LEO satellites do not change in this plane. Therefore, every LEO satellite can be represented by a two-dimensional coordinates(x, y) to shield the dynamic topology of LEO satellite network. Two-dimensional coordinates for all LEO satellites are set [20] and then routing strategy related to the two-dimensional coordinate is conducted. Roth et al. [21] proposed a routing algorithm based on geographic address identifiers in layer 2 of the communication stack, which can flexibly select an address resolution scheme. This method greatly reduces end-to-end delay and reroute rate.

QoS routing protocol is proposed to meet the needs of some multimedia applications, including audio applications, video applications, delay sensitive applications, packet drop sensitive applications. Bayhan et al. proposed ARPQ [22] (Adaptive Routing Protocol for Quality of Service) routing protocol, which is mainly suitable for delay-sensitive applications, such as VOIP (voiceover internet protocol). They firstly set a threshold and then classify incoming packets for transmission. When the transmission delay of packets is less than this threshold, the packets are directly transmitted to the destination node in the LEO satellite layer. If transmission delay is greater than this threshold, the transmission is carried out in the MEO layer. The disadvantage is its poor ability to predict congestion. Zhou et al. proposed HDRP [23] (Hierarchical and Distributed QoS Routing Protocol) based on the satellite group and group management. Comparing with MLSR and SGRP, it does not adopt virtual node strategy to shield the dynamics of LEO satellite network, but uses the actual satellite location to define the group members. In order to reduce the number of snapshots, they further propose a snapshot merge strategy. With the application of heuristic algorithm, the routing performance of QoS routing protocols have been greatly improved. Long [24] et al. proposed optimizing the routing strategy by using ant colony algorithm and genetic algorithm to improve the performance of satellite networks.

The traffic balancing routing protocol is also a hot issue. It can improve total throughput and reduce packet drop rate. Because satellite network topology is predictable and cyclical, traffic congestion can be predicted and the routing strategy can be adjusted in advance. Nishiyama et al. [25] have proposed a QoS routing protocol to balance traffic and traffic is divided into three types: delay sensitive traffic, video application traffic, and general transmission traffic. The time delay sensitive traffic has the highest priority. The video application traffic is transmitted in LEO satellites and general transmission traffic is transmitted in GEO satellites. The disadvantage is that the MEO layer can be congested. In order to solve this problem, Kawamoto et al. [26] proposed that traffic is sent through several MEO satellites. What’s more, Nishiyama et al. [27] further theoretically calculated the traffic deflection threshold. When the traffic exceeds this threshold, it is transmitted to MEO satellites. Otherwise, it is transmitted directly at the LEO layer.

## 3. Fuzzy sets Theory

### 3.1. Hesitant Fuzzy Set

The HFS (Hesitant fuzzy set) method was put forward by Torra [28] and Torra and Narukawa [29]. This method can solve the problem of different types of elements in a given set, including some multi-objective optimization problems. The following two definitions of the hesitant fuzzy element are given below.

**Definition** **1.***Refs.* [28,29] *Assuming that*
$X=\left\{x_{1},x_{2},x_{3},\ldots ,x_{m}\right\}$
*is a nonempty set, the HFS is a function of the subset from X to [0, 1]. For convenience, the HFS is expressed as*
$$A=\{<x,h_{A}\left(x\right)>|x\in X\}$$*where,*
$h_{A}\left(x\right)$
*is a set of some numbers in the interval [0, 1] and it indicates the possibility that element x belongs to the set A. At this time, we call*
$h_{A}\left(x\right)$
*a hesitant fuzzy element.*

**Definition** **2.***Assuming that* $X=\left\{x_{1},x_{2},x_{3},\ldots ,x_{m}\right\}$*is a nonempty set and*h*is a hesitant fuzzy element, we call*$s\left(h\right)=\frac{1}{l_{h}}\underset{y\in h}{\sum }y$*is the score of*h*. Where,*$l_{h}$*is an element of h. For two HFSs*$h_{1}$*and*$h_{2}$*, if*$s\left(h_{1}\right)>s\left(h_{2}\right)$*, then*$h_{1}>h_{2}$*; if*$s\left(h_{1}\right)=s\left(h_{2}\right)$*, then*$h_{1}=h_{2}$.*For hesitant fuzzy elements*h*,*$h_{1}$*,*$h_{2}$*,**Refs.* [27,28] *defined some operations:*(1)$$h^{\lambda }=\cup_{𝛾\in h}\left\{{𝛾}^{\lambda }\right\}$$
(2)$$\lambda h={\cup^{}}_{𝛾\in h}\left\{1-{\left(1-𝛾\right)}^{\lambda }\right\}$$
(3)$$h_{1}⨁h_{2}={\cup^{}}_{{𝛾}_{1}\in h_{1},{𝛾}_{2}\in h_{2}}\left\{{𝛾}_{1}+{𝛾}_{2}-{𝛾}_{1}{𝛾}_{2}\right\}$$
(4)$$h_{1}⨂h_{2}={\cup^{}}_{{𝛾}_{1}\in h_{1},{𝛾}_{2}\in h_{2}}\left\{{𝛾}_{1}{𝛾}_{2}\right\}$$

### 3.2. Interval Type-2 Fuzzy Set

The T1FSs (type-1 fuzzy sets) is a method proposed in 1965. The value of elements in Type-1 is represented by a real value in [0, 1]. Type-2 fuzzy sets have the characteristics of primary and secondary membership. In this part, we introduce the definition of Type-2 fuzzy sets and the definition of Interval Type-2 fuzzy sets. Trapezoidal type-1 fuzzy set can be represented by $\tilde{A}=\left\{p_{1},p_{2},p_{3},p_{4};P_{1}\left(A\right),P_{2}\left(A\right)\right\}$, where $p_{1},p_{2},p_{3},p_{4}$ are members of the set X. $P_{1}\left(A\right),P_{2}\left(A\right)$ are the non-zero values of the mapping function.

**Definition** **3.**
*Type-2 fuzzy set is an extension of the T1FSs. The membership degree of Type-2 fuzzy set is a Type-1 fuzzy set. Assuming that*

$\tilde{\tilde{A}}$

*is the Type-2 fuzzy set of set X, then the member relationship function*

${ս}_{\tilde{\tilde{A}}}$

*is satisfied with the following conditions:*

$$\tilde{\tilde{A}}=\{\left(\left(x,u\right),{ս}_{\tilde{\tilde{A}}}\left(x,u\right)\right)|\forall x\in X,\forall u\in J_{x}\subseteq \left[0,1\right],0\leq {ս}_{\tilde{\tilde{A}}}\left(x,u\right)\leq 1\}$$

*where*

$J_{x}$

*represents the interval of [0, 1], so the Type-2 fuzzy set*

$\tilde{\tilde{A}}$

*also can be represented as follows:*

$$\tilde{\tilde{A}}=\overset{}{\underset{x\in X}{\int }}\overset{}{\underset{u\in J_{x}}{\int }}{ս}_{\tilde{\tilde{A}}}\left(x,u\right)|\left(x,u\right)$$



**Definition** **4.**
*Assuming that*

$\tilde{\tilde{A}}$

*is the Type-2 fuzzy set of set X and the member relationship function is*

${ս}_{\tilde{\tilde{A}}}$

*, if all*

${ս}_{\tilde{\tilde{A}}}\left(x,u\right)$

*= 1, then*

$\tilde{\tilde{A}}$

*is called the Interval Type-2 fuzzy set of set X. Interval Type-2 fuzzy set is a special fuzzy set and can be represented by:*

$$\tilde{\tilde{A}}=\overset{}{\underset{x\in X}{\int }}\overset{}{\underset{u\in J_{x}}{\int }}1|\left(x,u\right)$$

*where*

$J_{x}$

*represents the interval of [0, 1].*


**Definition** **5.***The membership function of Type-1 fuzzy set is composed of the upper and the lower membership functions of Interval Type-2 fuzzy set. We use interval Type-2 fuzzy set to solve multi-QoS routing problem. The upper and the lower membership functions of Interval Type-2 fuzzy set are defined as:*$${\tilde{\tilde{A}}}_{i}=\left({\tilde{A}}_{i}^{u},{\tilde{A}}_{i}^{l}\right)=\left(p_{i1}^{u},p_{i2}^{u},p_{i3}^{u},p_{i4}^{u};P_{1}\left({\tilde{A}}_{i}^{u}\right),P_{2}\left({\tilde{A}}_{i}^{u}\right)\right),\left(p_{i1}^{l},p_{i2}^{l},p_{i3}^{l},p_{i4}^{l};P_{1}\left({\tilde{A}}_{i}^{l}\right),P_{2}\left({\tilde{A}}_{i}^{l}\right)\right)$$*where*${\tilde{A}}_{i}^{u},{\tilde{A}}_{i}^{l}$*denote the upper and the lower membership functions and they are T1FSs;*$p_{i1}^{u},p_{i2}^{u},p_{i3}^{u},p_{i4}^{u}$*and*$p_{i1}^{l},p_{i2}^{l},p_{i3}^{l},p_{i4}^{l}$*denote the reference points of interval Type-2 fuzzy set*$\;\tilde{\tilde{A}}$. $P_{1}\left({\tilde{A}}_{i}^{u}\right),P_{2}\left({\tilde{A}}_{i}^{u}\right)$*denote the value of the upper membership function of*$p_{i2}^{u}$*and*$p_{i3}^{u}$. $P_{1}\left({\tilde{A}}_{i}^{l}\right),P_{2}\left({\tilde{A}}_{i}^{l}\right)$*denote the value of the lower membership function of*$p_{i2}^{l},p_{i3}^{l}$*. Where*1≪i≪n*, n is the number of elements of the x set.*$P_{1}\left({\tilde{A}}_{i}^{u}\right),P_{2}\left({\tilde{A}}_{i}^{u}\right),P_{1}\left({\tilde{A}}_{i}^{l}\right),P_{2}\left({\tilde{A}}_{i}^{l}\right)$*both are the value of [0, 1].*

**Definition** **6.**
*According to the above definitions, we can define the following operations:*



$${\tilde{\tilde{A}}}_{1}=\left({\tilde{A}}_{1}^{u},{\tilde{A}}_{1}^{l}\right)=\left(p_{11}^{u},p_{12}^{u},p_{13}^{u},p_{14}^{u};P_{1}\left({\tilde{A}}_{1}^{u}\right),P_{2}\left({\tilde{A}}_{1}^{u}\right)\right),\left(p_{11}^{l},p_{12}^{l},p_{13}^{l},p_{14}^{l};P_{1}\left({\tilde{A}}_{1}^{l}\right),P_{2}\left({\tilde{A}}_{1}^{l}\right)\right)$$



$${\tilde{\tilde{A}}}_{2}=\left({\tilde{A}}_{2}^{u},{\tilde{A}}_{2}^{l}\right)=\left(p_{21}^{u},p_{22}^{u},p_{23}^{u},p_{24}^{u};P_{1}\left({\tilde{A}}_{2}^{u}\right),P_{2}\left({\tilde{A}}_{2}^{u}\right)\right),\left(p_{21}^{l},p_{22}^{l},p_{23}^{l},p_{24}^{l};P_{1}\left({\tilde{A}}_{2}^{l}\right),P_{2}\left({\tilde{A}}_{2}^{l}\right)\right)\;$$



$$\begin{array}{l} {\tilde{\tilde{A}}}_{1}⨁{\tilde{\tilde{A}}}_{2}=({\tilde{A}}_{1}^{u},{\tilde{A}}_{1}^{l})⨁\left({\tilde{A}}_{2}^{u},{\tilde{A}}_{2}^{l}\right)\\ \;=\{(p_{11}^{u}+p_{21}^{u},p_{12}^{u}+p_{22}^{u},p_{13}^{u}+p_{23}^{u},p_{14}^{u}\\ \;+p_{24}^{u};\min\left(P_{1}\left({\tilde{A}}_{1}^{u}\right),P_{1}\left({\tilde{A}}_{2}^{u}\right)\right),\min\left(P_{2}\left({\tilde{A}}_{1}^{u}\right),P_{2}\left({\tilde{A}}_{2}^{u}\right)\right)),(p_{11}^{l}+p_{21}^{l},p_{12}^{l}+p_{22}^{l},p_{13}^{l}+p_{23}^{l},p_{14}^{l}\\ \;+p_{24}^{l};\min\left(P_{1}\left({\tilde{A}}_{1}^{l}\right),P_{1}\left({\tilde{A}}_{2}^{l}\right)\right),\min\left(P_{2}\left({\tilde{A}}_{1}^{l}\right),P_{2}\left({\tilde{A}}_{2}^{l}\right)\right))\} \end{array}$$



$$\begin{array}{l} {\tilde{\tilde{A}}}_{1}⊝{\tilde{\tilde{A}}}_{2}=\left({\tilde{A}}_{1}^{u},{\tilde{A}}_{1}^{l}\right)⊝\left({\tilde{A}}_{2}^{u},{\tilde{A}}_{2}^{l}\right)\\ \;=\{(p_{11}^{u}-p_{21}^{u},p_{12}^{u}-p_{22}^{u},p_{13}^{u}-p_{23}^{u},p_{14}^{u}\\ \;-p_{24}^{u};\min\left(P_{1}\left({\tilde{A}}_{1}^{u}\right),P_{1}\left({\tilde{A}}_{2}^{u}\right)\right),\min\left(P_{2}\left({\tilde{A}}_{1}^{u}\right),P_{2}\left({\tilde{A}}_{2}^{u}\right)\right)),(p_{11}^{l}-p_{21}^{l},p_{12}^{l}-p_{22}^{l},p_{13}^{l}-p_{23}^{l},p_{14}^{l}\\ \;-p_{24}^{l};\min\left(P_{1}\left({\tilde{A}}_{1}^{l}\right),P_{1}\left({\tilde{A}}_{2}^{l}\right)\right),\min\left(P_{2}\left({\tilde{A}}_{1}^{l}\right),P_{2}\left({\tilde{A}}_{2}^{l}\right)\right))\} \end{array}$$



$$\begin{array}{l} {\tilde{\tilde{A}}}_{1}⨂{\tilde{\tilde{A}}}_{2}=\left({\tilde{A}}_{1}^{u},{\tilde{A}}_{1}^{l}\right)⨂\left({\tilde{A}}_{2}^{u},{\tilde{A}}_{2}^{l}\right)\\ \;=\{(p_{11}^{u}\times p_{21}^{u},p_{12}^{u}\times p_{22}^{u},p_{13}^{u}\times p_{23}^{u},p_{14}^{u}\\ \;\times p_{24}^{u};\min\left(P_{1}\left({\tilde{A}}_{1}^{u}\right),P_{1}\left({\tilde{A}}_{2}^{u}\right)\right),\min\left(P_{2}\left({\tilde{A}}_{1}^{u}\right),P_{2}\left({\tilde{A}}_{2}^{u}\right)\right)),(p_{11}^{l}\times p_{21}^{l},p_{12}^{l}\times p_{22}^{l},p_{13}^{l}\times p_{23}^{l},p_{14}^{l}\\ \;\times p_{24}^{l};\min\left(P_{1}\left({\tilde{A}}_{1}^{l}\right),P_{1}\left({\tilde{A}}_{2}^{l}\right)\right),\min\left(P_{2}\left({\tilde{A}}_{1}^{l}\right),P_{2}\left({\tilde{A}}_{2}^{l}\right)\right))\} \end{array}$$



*For the trapezoidal interval type-2 fuzzy set*

${\tilde{\tilde{A}}}_{1}$

*, there are the following operations:*

$${\tilde{\tilde{A}}}_{1}=\left({\tilde{A}}_{1}^{u},{\tilde{A}}_{1}^{l}\right)=\left(p_{11}^{u},p_{12}^{u},p_{13}^{u},p_{14}^{u};P_{1}\left({\tilde{A}}_{1}^{u}\right),P_{2}\left({\tilde{A}}_{1}^{u}\right)\right),\left(p_{11}^{l},p_{12}^{l},p_{13}^{l},p_{14}^{l};P_{1}\left({\tilde{A}}_{1}^{l}\right),P_{2}\left({\tilde{A}}_{1}^{l}\right)\right)$$


$$k{\tilde{\tilde{A}}}_{1}=\left(kp_{11}^{u},kp_{12}^{u},kp_{13}^{u},kp_{14}^{u};P_{1}\left({\tilde{A}}_{1}^{u}\right),P_{2}\left({\tilde{A}}_{1}^{u}\right)\right),\left(kp_{11}^{l},kp_{12}^{l},kp_{13}^{l},kp_{14}^{l};P_{1}\left({\tilde{A}}_{1}^{l}\right),P_{2}\left({\tilde{A}}_{1}^{l}\right)\right)$$



*where* k>0.

### 3.3. Interval Type-2 Hesitant Fuzzy Set

In order to simplify the calculation, [29] put forward interval type-2 hesitant fuzzy set based on the interval type-2 fuzzy set.

**Definition** **7.**
*Assuming that X is a fixed set, interval type-2 hesitant fuzzy set is a subset of interval type-2 fuzzy set. It can be expressed mathematically as:*



$$E=\{<x,{\tilde{h}}_{E}\left(x\right)>|x\in X\}$$


*where*${\tilde{h}}_{E}\left(x\right)$*is the element of [0, 1] and denote the probability of*x∈X. *The formula is as follows:*$${\tilde{h}}_{E}\left(x\right)=\tilde{h}=\{{\tilde{A}}_{i}\in \tilde{h}|{\tilde{A}}_{i}=\left(p_{i1}^{u},p_{i2}^{u},p_{i3}^{u},p_{i4}^{u};P_{1}\left({\tilde{A}}_{i}^{u}\right),P_{2}\left({\tilde{A}}_{i}^{u}\right)\right),\left(p_{i1}^{l},p_{i2}^{l},p_{i3}^{l},p_{i4}^{l};P_{1}\left({\tilde{A}}_{i}^{l}\right),P_{2}\left({\tilde{A}}_{i}^{l}\right)\right)\}$$


*If*

$${\tilde{h}}_{1}=\{{\tilde{A}}_{1}\in \tilde{h}|{\tilde{A}}_{1}=\left(p_{11}^{u},p_{12}^{u},p_{13}^{u},p_{14}^{u};P_{1}\left({\tilde{A}}_{1}^{u}\right),P_{2}\left({\tilde{A}}_{1}^{u}\right)\right),\left(p_{11}^{l},p_{12}^{l},p_{13}^{l},p_{14}^{l};P_{1}\left({\tilde{A}}_{1}^{l}\right),P_{2}\left({\tilde{A}}_{1}^{l}\right)\right)\}$$


$${\tilde{h}}_{2}=\{{\tilde{A}}_{2}\in \tilde{h}|{\tilde{A}}_{2}=\left(p_{21}^{u},p_{22}^{u},p_{23}^{u},p_{24}^{u};P_{1}\left({\tilde{A}}_{2}^{u}\right),P_{2}\left({\tilde{A}}_{2}^{u}\right)\right),\left(p_{21}^{l},p_{22}^{l},p_{23}^{l},p_{24}^{l};P_{1}\left({\tilde{A}}_{2}^{l}\right),P_{2}\left({\tilde{A}}_{2}^{l}\right)\right)\}$$




*Then*

$${\tilde{h}}_{1}^{\lambda }={\cup^{}}_{{\tilde{A}}_{1}\in {\tilde{h}}_{1}}=\{\left(k^{-1}(\lambda kp_{11}^{u}),k^{-1}(\lambda kp_{12}^{u}),k^{-1}(\lambda kp_{13}^{u}),k^{-1}(\lambda kp_{14}^{u});P_{1}\left({\tilde{A}}_{1}^{u}\right),P_{2}\left({\tilde{A}}_{1}^{u}\right)\right),\;\left(k^{-1}(\lambda kp_{11}^{l}),k^{-1}(\lambda kp_{12}^{l}),k^{-1}(\lambda kp_{13}^{l}),k^{-1}(\lambda kp_{14}^{l});P_{1}\left({\tilde{A}}_{1}^{l}\right),P_{2}\left({\tilde{A}}_{1}^{l}\right)\right)\}$$


$$\lambda {\tilde{h}}_{1}={\cup^{}}_{{\tilde{A}}_{1}\in {\tilde{h}}_{1}}=\{\left(q^{-1}(\lambda qp_{11}^{u}),q^{-1}(\lambda qp_{12}^{u}),q^{-1}(\lambda qp_{13}^{u}),q^{-1}(\lambda qp_{14}^{u});P_{1}\left({\tilde{A}}_{1}^{u}\right),P_{2}\left({\tilde{A}}_{1}^{u}\right)\right),\;\left(q^{-1}(\lambda qp_{11}^{l}),q^{-1}(\lambda qp_{12}^{l}),q^{-1}(\lambda qp_{13}^{l}),q^{-1}(\lambda qp_{14}^{l});P_{1}\left({\tilde{A}}_{1}^{l}\right),P_{2}\left({\tilde{A}}_{1}^{l}\right)\right)\}$$


$$\begin{array}{l} {\tilde{h}}_{1}⊕{\tilde{h}}_{2}={\cup^{}}_{{\tilde{A}}_{1}\in {\tilde{h}}_{1},{\tilde{A}}_{2}\in {\tilde{h}}_{2}}\\ \;=\{(k^{-1}\left(kp_{11}^{u}+kp_{21}^{u}\right),k^{-1}(kp_{12}^{u}+kp_{22}^{u}),k^{-1}\left(kp_{13}^{u}+kp_{23}^{u}\right),k^{-1}(kp_{14}^{u}\\ \;+kp_{24}^{u});\min\left(P_{1}\left({\tilde{A}}_{1}^{u}\right),P_{1}\left({\tilde{A}}_{2}^{u}\right)\right),\min\left(P_{2}\left({\tilde{A}}_{1}^{u}\right),P_{2}\left({\tilde{A}}_{2}^{u}\right)\right)),(k^{-1}\left(kp_{11}^{l}+kp_{21}^{l}\right),k^{-1}(kp_{12}^{l}\\ \;+kp_{22}^{l}),k^{-1}\left(kp_{13}^{l}+kp_{23}^{l}\right),k^{-1}\left(kp_{14}^{l}+kp_{24}^{l}\right);\min\left(P_{1}\left({\tilde{A}}_{1}^{l}\right),P_{1}\left({\tilde{A}}_{2}^{l}\right)\right),\min\left(P_{2}\left({\tilde{A}}_{1}^{l}\right),P_{2}\left({\tilde{A}}_{2}^{l}\right)\right)) \end{array}$$


$$\begin{array}{l} {\tilde{h}}_{1}⨂{\tilde{h}}_{2}={\cup^{}}_{{\tilde{A}}_{1}\in {\tilde{h}}_{1},{\tilde{A}}_{2}\in {\tilde{h}}_{2}}\\ \;=\{(k^{-1}\left(kp_{11}^{u}+kp_{21}^{u}\right),k^{-1}(kp_{12}^{u}+kp_{22}^{u}),k^{-1}\left(kp_{13}^{u}+kp_{23}^{u}\right),k^{-1}(kp_{14}^{u}\\ \;+kp_{24}^{u});\min\left(P_{1}\left({\tilde{A}}_{1}^{u}\right),P_{1}\left({\tilde{A}}_{2}^{u}\right)\right),\min\left(P_{2}\left({\tilde{A}}_{1}^{u}\right),P_{2}\left({\tilde{A}}_{2}^{u}\right)\right)),(k^{-1}\left(kp_{11}^{l}+kp_{21}^{l}\right),k^{-1}(kp_{12}^{l}\\ \;+kp_{22}^{l}),k^{-1}\left(kp_{13}^{l}+kp_{23}^{l}\right),k^{-1}\left(kp_{14}^{l}+kp_{24}^{l}\right);\min\left(P_{1}\left({\tilde{A}}_{1}^{l}\right),P_{1}\left({\tilde{A}}_{2}^{l}\right)\right),\min\left(P_{2}\left({\tilde{A}}_{1}^{l}\right),P_{2}\left({\tilde{A}}_{2}^{l}\right)\right)) \end{array}$$



## 4. Multi-QoS Routing Model

### 4.1. Software-Defined Satellite Network

As an important part of the next generation of 6G mobile communication, LEO satellites communication has been playing an increasingly important role in disaster relief and geological survey. In recent years, the user’s number and data of LEO satellite network have increased dramatically.

The on-board processing and storage capacity of LEO satellites are very limited and they can’t meet the need of storing and processing satellite connection relationships.

In this section, we modeled the routing in the air-space-ground integrated network. The SDSN architecture is as follows:

As shown in Figure 1, the data plane is composed of LEO satellites and portable satellite terminals (PSTs). The control plane is composed of ground stations, controllers and location servers. The controller generates and sends the routing instruction to the LEO satellites via a SNOF (satellite network OpenFlow) channel. The LEO satellites communicate with each other through ISLs.

### 4.2. Routing Model

We assume that the LEO satellite constellation is the iridium constellation, which consists of M*N satellites, where M denotes the number of LEO satellites in each plane, N denotes the number of LEO satellites planes and the radians between two adjacent satellites is 2π/N. For each LEO satellite, there are four ISLs except the polar or reverse seam according to satellite visible light communications [30]. If the number of satellites is more than four, that would be better. But for simplicity, we assume there are only four ISLs. Each LEO satellite repeater has a send buffer and has no receive buffer.

The two ends of inter-satellite link can transmit data to each other and four directions of the four links are regarded as the four interfaces of the router. When a satellite receives a packet, the packet is placed in the sending buffer based on the routing algorithm and the packet that stores the forwarding is waiting for the buffer.

As we can see in Figure 2, the sender satellite is LEO11 and receiver satellite is LEO55. When a packet is transmitted to LEO11, LEO11 puts it to a QoS queue in send buffer according to the application type. According to the proposed routing algorithm, LEO11 selects the next hop and sends the packet. Until the packet arrives in LEO55, receiver satellite(LEO55) gets the packet.

In order to meet the requirements of multi-QoS routing targets, the queues in sender buffer are classified according to the application type. In this paper, the application service type is set to four kinds, including time-delay sensitive application, video application, packet loss sensitive application and no special QoS requirements application. The congestion level of each interface depends on the sending of packets from all QoS queues. Figure 3 depicts the queue model.

## 5. Proposed Methodology

In this section, we will introduce the proposed multi-QoS routing algorithm that uses the Interval type-2 fuzzy set to make fuzzy decisions.

In general, the steps used to make fuzzy decisions that uses Interval type-2 fuzzy set are the following:

Step1: Formulate the multi criteria decision (Multi-QoS) solving problem by determining criteria set as $B=\left\{m_{1},m_{2},m_{3},\ldots ,m_{n}\right\}$ with the criteria weight vector W = {$w_{1}$,$\;w_{2}$, …,$\;w_{n}$} and $\overset{n}{\underset{j=1}{\sum }}w_{j}=1$.

Step2: Determine the linguistic term set, semantic and linguistic expressions. Define $V^{g}$ is the context free expression of linguistic term set and S is a linguistic term set and S = $\left\{\;S_{0},S_{1},S_{2},S_{3},\cdots ,S_{d}\right\}$. There is a rule: if i≪j, then $S_{i}\ll S_{j}$. And then define the operation: $\max\left(S_{i},S_{j}\right)=S_{j}$, $\min\left(S_{i},S_{j}\right)=S_{i}$, $V^{g}=\left\{at\;least,\;at\;most,\;between,\;is\;and\;S_{0},S_{1},S_{2},S_{3},\cdots ,S_{d}\right\}$. The rule is: R={at least≔≫”; at most:=≪; between≔<⋯<; is≔”=“}.

Step3: Assume that there are k experts and collect the negative and positive comments of all *k* experts on all criteria. In order to describe interval type-2 fuzzy set, define the evaluations are in $\left[r_{ij}^{-},r_{ij}^{+}\right]$, $r_{ij}^{-},r_{ij}^{+}$ respectively represent the minimum of negative evaluation value and maximum of positive evaluation value of the *i* expert on the *j* goal.

Step4: The data given by experts is de-fuzzed. The corresponding value is given in Table 1.

From Table 1, the data given by experts can be expressed by interval type-2 fuzzy set:$$\tilde{h}=\{{\tilde{A}}_{i}\in \tilde{h}|{\tilde{A}}_{i}=\left(p_{i1}^{u},p_{i2}^{u},p_{i3}^{u},p_{i4}^{u};P_{1}\left({\tilde{A}}_{i}^{u}\right),P_{2}\left({\tilde{A}}_{i}^{u}\right)\right),\left(p_{i1}^{l},p_{i2}^{l},p_{i3}^{l},p_{i4}^{l};P_{1}\left({\tilde{A}}_{i}^{l}\right),P_{2}\left({\tilde{A}}_{i}^{l}\right)\right)\}.$$
where ${\tilde{h}}_{ij}$ is interval type-2 fuzzy set, *i* denotes scheme and *j* denotes criterion.

Step5: According to all target demand data, evaluate all schemes based on ($w_{j}1\leq j\leq n$):
$$\begin{array}{l} {\tilde{h}}_{i}=\overset{n}{\underset{j=1}{\sum }}w_{j}{\tilde{h}}_{ij}\left(\text{where}\;i=1,2,3,\ldots ,\text{m}\right)\\ \;={\cup^{}}_{{\tilde{A}}_{i1}\in {\tilde{h}}_{i1},{\tilde{A}}_{i2}\in {\tilde{h}}_{i2},\ldots ,{\tilde{A}}_{in}\in {\tilde{h}}_{in}}\{\left(\begin{array}{c} k^{-1}(\overset{n}{\underset{j=1}{\sum }}w_{j}k(p_{ij1}^{u})),k^{-1}(\overset{n}{\underset{j=1}{\sum }}w_{j}k(p_{ij2}^{u})),k^{-1}(\overset{n}{\underset{j=1}{\sum }}w_{j}k(p_{ij3}^{u})),\\ k^{-1}(\overset{n}{\underset{j=1}{\sum }}w_{j}k(p_{ij4}^{u}));min_{j}\left(P_{1}\left({\tilde{A}}_{ij}^{u}\right)\right),min_{j}\left(P_{2}\left({\tilde{A}}_{ij}^{u}\right)\right) \end{array}\right), \end{array}\;\left(\begin{array}{c} k^{-1}(\overset{n}{\underset{j=1}{\sum }}w_{j}k(p_{ij1}^{l})),k^{-1}(\overset{n}{\underset{j=1}{\sum }}w_{j}k(p_{ij2}^{l})),k^{-1}(\overset{n}{\underset{j=1}{\sum }}w_{j}k(p_{ij3}^{l})),\\ k^{-1}(\overset{n}{\underset{j=1}{\sum }}w_{j}k(p_{ij4}^{l}));min_{j}\left(P_{1}\left({\tilde{A}}_{ij}^{l}\right)\right),min_{j}\left(P_{2}\left({\tilde{A}}_{ij}^{l}\right)\right) \end{array}\right)\}$$

Step6: Calculate the score $s\left({\tilde{h}}_{ij}\right)$(i=1,2,3,…,m) for each scheme. If $\tilde{h}=\{{\tilde{A}}_{i}\in \tilde{h}|{\tilde{A}}_{i}=\left(p_{i1}^{u},p_{i2}^{u},p_{i3}^{u},p_{i4}^{u};P_{1}\left({\tilde{A}}_{i}^{u}\right),P_{2}\left({\tilde{A}}_{i}^{u}\right)\right),\left(p_{i1}^{l},p_{i2}^{l},p_{i3}^{l},p_{i4}^{l};P_{1}\left({\tilde{A}}_{i}^{l}\right),P_{2}\left({\tilde{A}}_{i}^{l}\right)\right)\}$ is an interval type-2 fuzzy set, then score function $s\left(\tilde{h}\right)=\frac{1}{\left|\tilde{h}\right|}\underset{\tilde{A}\epsilon \tilde{h}}{\sum }s\left(\tilde{A}\right)$ = $\frac{1}{\left|\tilde{h}\right|}\underset{\tilde{A}\epsilon \tilde{h}}{\sum }\left[\frac{p_{1}^{u}+p_{4}^{u}}{2}+\frac{P_{1}\left({\tilde{A}}^{u}\right)+P_{2}\left({\tilde{A}}^{u}\right)+P_{1}\left({\tilde{A}}^{l}\right)+P_{2}\left({\tilde{A}}^{l}\right)}{4}\right]\times \left(\frac{p_{1}^{u}+p_{2}^{u}+p_{3}^{u}+p_{4}^{u}+p_{1}^{l}+p_{2}^{l}+p_{3}^{l}+p_{4}^{l}}{8}\right)$. Where $\left|\tilde{h}\right|$ is the number of elements of interval type-2 fuzzy set $\tilde{h}$. And the value of score function $s\left(\tilde{h}\right)$ is in [0, 1]. For $s\left(\tilde{h}\right)$, if $s\left({\tilde{h}}_{1}\right)\gg s\left({\tilde{h}}_{2}\right)$, then ${\tilde{h}}_{1}\gg {\tilde{h}}_{2}$.

The calculation of Step5 and Step6 is based on the formulas in Section 3.

Step7: The dominance matrix is established on the dominance relationship. Using the score function, the interval number of each target is calculated. The boundary of the interval number is optimistic evaluation and pessimistic evaluation respectively. If $I_{1}=\left[a_{1},b_{1}\right]$, $I_{2}=\left[a_{2},b_{2}\right]$, then relationship of $I_{1}$ and $I_{2}$($I_{1}>I_{2}$)can be calculated as:$$R\left(I_{1}>I_{2}\right)=\frac{\max\left(0,b_{1}-a_{2}\right)-\max\left(0,a_{1}-b_{2}\right)}{\left(b_{1}-a_{1}\right)+\left(b_{2}-a_{2}\right)}$$

Similarly, $R\left(I_{2}>I_{1}\right)$ can be calculated. $R\left(I_{1}>I_{2}\right)+R\left(I_{2}>I_{1}\right)=1$. Specially, when $a_{1}=a_{2}$ and $b_{1}=b_{2}$, $R\left(I_{1}>I_{2}\right)=R\left(I_{2}>I_{1}\right)$=0.5. If we want to compare how much $I_{1}$ is greater than $I_{2}$, the following equation can be conducted:$$DM_{12}=\max\left(0,R\left(I_{1}>I_{2}\right)-R\left(I_{2}>I_{1}\right)\right).$$

Step8: Use non dominance rule that Rodriguez et al. (2012) put forward:$$NDM_{i}=\left|\min\left(\left(1-DM_{1}\right),\left(1-DM_{2}\right),\ldots ,\left(1-DM_{n}\right)\right)\right|(n\neq i).$$

Step9: Normalization.
$$NDM_{i}=\frac{NDM_{i}}{\sum_{i=1}^{n}NDM_{i}}$$

### 5.1. Linguistic Description and Rule Base

In this section, we define the linguistic description and rule:

Traffic Class A: The flow of typical delay-sensitive applications, such as some real-time communication and audio applications.

Traffic Class B: The flow generated by the typical video applications and requiring throughput assurance.

Traffic Class C: The flow generated by the packet-loss sensitive applications and needing to keep a small packet-loss rate.

Traffic Class D: The flow generated by the applications that has no special QoS requirements.

When a flow is sent to a LEO satellite, multi-QoS requirements are discussed: delay, throughput and packet loss rate. The criteria can fully denote current network state at the moment. However, the criteria cannot be calculated instantaneously and other measures are needed to express these three QoS requirements.

In this paper, we use the queuing length of traffic class A ($L_{A}$), the queuing length of traffic class B ($L_{B}$), the queuing length of traffic class C ($L_{C}$) and the queuing length of traffic class D ($L_{D}$) as the input of proposed routing algorithm to the indirectly describe of the current network state. The queuing length of traffic class can infer the congestion degree of ISLs. For each queuing length of traffic class, there are eight kinds of evaluation: “AL”, “VL”, “ML”, “M”, “MH”, “H”, “VH” and “AH”. Their corresponding values is given by Table 1.

### 5.2. Routing Strategy

(1) Control plane: In this paper, we use virtual topology strategy to shield the dynamic topology of LEO satellite network. The satellite network topology is unchanged in each slot and the satellite movement is cyclical. In each hot, location servers send control information and update routing tables of LEO satellites through GEO satellites transmitting control information. When updating routing tables, the proposed interval type-2 fuzzy set decision routing algorithm is adopted based on the information received by the former network.

(2) Data plane: After location servers receive the packets from users, add the application attributes in the header of the packets to distinguish the kind of traffic class. When a location server send a packet to a LEO satellite, the LEO satellite put it into corresponding packet queues based on the properties of the packet header. The packets of traffic Class A is put in $Q_{1}$ queue. The packets of traffic Class B is put in $Q_{2}$ queue. The packets of traffic Class C is put in $Q_{3}$ queue. The packets of traffic Class D is put in $Q_{4}$ queue. When packets are transmitted between LEO satellites, the sending satellites send packets to the next hop based on their latest routing table. The next hop puts packets into corresponding packet queues based on the properties of the packet header and transmit them according to the routing table until the transmission is completed.

(3) Routing algorithm: When sender send a packet, the packet is placed in the corresponding queue of the sending buffer. The corresponding port is selected according to the routing table. The update of the routing table is based on Algorithm 1.
**Algorithm 1.** Routing algorithm based on the Interval type-2 fuzzy set1: input: $L_{A}$, $L_{B}$, $L_{C}$, $L_{D}$
2: output: Routing Table
3: Initialize all computing period (Δt) and timer(t) in this algorithm.
4: Every LEO satellite collects the $L_{A}$, $L_{B}$, $L_{C}$, $L_{D}$ and sends them to GEO satellites.
5: Every GEO satellite sends packets ($L_{A}$, $L_{B}$, $L_{C}$, $L_{D}$) to ground servers.
6: The ground servers calculate the control information.
7: The ground servers send the control information to GEO satellites.
8: Every GEO satellite sends the control information to LEO satellites.
9: Every LEO satellite adjusts its transmission mode and updates its Routing Table. $\mathrm{In}\;\mathrm{the}\;\mathrm{case}\;\mathrm{of}\;\mathrm{step}6,\;\mathrm{the}\;\mathrm{procedure}\;\mathrm{of}\;\mathrm{fuzzing}\;\mathrm{the}\;\mathrm{network}\;\mathrm{state}\;\mathrm{information}\;(L_{A}$$,\;L_{B}$$,\;L_{C}$$,\;L_{D}$) not only take the form of expert ratings, but also can be defined by the membership function. In this paper, the fuzzy class number of each criteria is 8 and we divide the value range of each criteria into eight fuzzy classes by using the trapezoidal membership function method.

The trapezoidal membership functions are as follows:$$g^{1}(x)=\left\{ \begin{array}{l} 1\begin{array}{cc} & \begin{array}{cc} & \begin{array}{cc} & x\leq \bar{\lambda_{1}} \end{array} \end{array} \end{array}\\ \frac{\underline{\lambda_{2}}-x}{\underline{\lambda_{2}}-\bar{\lambda_{1}}}\bar{\lambda_{1}}<x\leq \underline{\lambda_{2}}\\ 0x\geq \underline{\lambda_{2}} \end{array}\right.$$
$$g^{k}(x)=\left\{ \begin{array}{l} \frac{x-\bar{\lambda_{k-1}}}{\underline{\lambda_{k}}-\bar{\lambda_{k-1}}}\begin{array}{cc} & \begin{array}{cc} & \begin{array}{cc} & x\in [\bar{\lambda_{k-1}},\underline{\lambda_{k}}] \end{array} \end{array} \end{array}\\ 1x\in [\underline{\lambda_{k}},\bar{\lambda_{k}}]\\ \frac{\underline{\lambda_{k+1}}-x}{\underline{\lambda_{k+1}}-\bar{\lambda_{k}}}x\in [\bar{\lambda_{k}},\underline{\lambda_{k+1}}]\\ 0\begin{array}{cccc} & & & \begin{array}{ccc} & & x\notin [\bar{\lambda_{k-1}},\underline{\lambda_{k+1}}] \end{array} \end{array} \end{array}\right.(k=2,3,4,5,6,7)$$
$$g^{8}(x)=\left\{ \begin{array}{l} 0\begin{array}{cc} & \begin{array}{cc} & \begin{array}{cc} & x\leq \bar{\lambda_{7}} \end{array} \end{array} \end{array}\\ \frac{x-\bar{\lambda_{7}}}{\underline{\lambda_{8}}-\bar{\lambda_{7}}}x\in [\bar{\lambda_{7}},\underline{\lambda_{8}}]\\ 1\begin{array}{cccc} & & & \begin{array}{ccc} & & x\geq \underline{\lambda_{8}} \end{array} \end{array} \end{array}\right.$$

The membership functions are in Table 2.

As the smaller the queuing length of traffic class is, the better, it belongs to the negative criteria. The evaluation of every criteria includes “AL” “VL” “ML” “M” “MH” “H” “VH” “AH” and the values of parameters are in Table 3.

After the evaluation of every criteria is obtained, find corresponding IT2HFNs. According to all the queue data and weight coefficient, evaluate all schemes. In order to ensure the normal use of all applications, all weight coefficient is set as $\;\frac{1}{4}$ ($w_{1}=w_{2}=w_{3}=w_{4}=\frac{1}{4}$). Calculate the score $s\left({\tilde{h}}_{ij}\right)$ of every LEO satellite and the dominance matrix is established on the dominance relationship. The update information of routing table is obtained.

## 6. Experiments and Results

In this section, the environment and establishment of the experiment are introduced firstly. And then the results and analysis of delay, throughput and packet loss rate are given.

### 6.1. Experiments Establishment

In order to simulate the experimental environment of satellite network, we use the ns2 simulator to simulate the experiment. All simulations about LEO satellites have been carried out on a typical LEO satellite constellation named Iridium. In this paper, the effectiveness of the algorithm in the topology of satellite network is verified. 30 stations are randomly distributed on the earth. The longitude and latitude range of stations are [−180°,180°] and [0°,90°] respectively. The detailed parameters of Iridium are shown in Table 4.

The buffer queue size on each node is set to 100 packets and the size of each packet is set to 1 KB. We use 1200 ON/OFF streams and the ON/OFF period of each flow follows the Pareto distribution with the shape parameter of 1.5. The average burst time and idle time are set to 500 ms. The range of individual data transmission rate is from 0.3 Mbps to 0.75 Mbps. The capacity of inter-satellite link and satellite- terrestrial link are set to 25 Mbps.

To evaluate end-to-end delay, the flow distribution model is shown in Table 5.

Distribution of traffic flows is determined by the density of population. Satellite traffic over densely populated cities, such as cities in North America, Europe and East Asia, is large, resulting in large queuing delay. The satellite traffic over sparsely populated areas, such as oceans and deserts, is small and the queuing time is small.

For simplicity, we only simulate the data plane (LEO satellite constellation). The control plane is only used to calculate the routing table, which can be simulated during the implementation of the routing algorithm. ELB and DSP algorithms are two classical satellite network routing algorithms and researchers have compared their proposed algorithms with them in many papers. we compare routing algorithm based on the Interval type-2 fuzzy set proposed in this paper with the ELB routing algorithm over DSP and the centralized DSP routing algorithm in average delay, total throughput and packet drop rate

### 6.2. Evaluation Indicators

The evaluation indicators in the experiment are average delay, total throughput and packet drop rate.

In SDSN, the average delay refers to a certain period of time when packets are sent from senders to receivers after it is transmitted through terrestrial network and satellite network. Assuming that the number of ISLs is N, the propagation delay of an inter-satellite link is $T_{k}$(k=1,2,3,…,n) and the propagation delay of a satellite-ground link is $T_{d}$. Transmission delay of each node is $T_{tran,a}(a=1,2,3,\ldots ,n,n+1,n+2)$, processing delay of each node is $T_{pro,b}(b=1,2,3,\ldots ,n,n+1,n+2)$ and the queuing delay of each node is $T_{queue,c}(c=1,2,3,\ldots ,n,n+1,n+2)$. Then average delay in SDSN is:$$T=\sum_{k=1}^{n}T_{k}+T_{d}+\sum_{a=1}^{n+2}T_{tran,a}+\sum_{b=1}^{n+2}T_{pro,b}+\sum_{c=1}^{n+2}T_{queue,c}$$

Total throughput is the rate of traffic arriving at the destination, which can be calculated as total number of packets arriving at destination terminal divided by the simulation time. Assuming that total number of packets sent from sending terminal to destination terminal is P and the simulation time is $T_{1}$, then total throughput is $\frac{P}{T_{1}}$.

Packet drop rate is calculated by packets sent and packets received. Assuming that total number of packets sent by sending node is $P_{send}$ and total number of packets received by receiving node is $P_{receive}$, packet drop rate can be calculated as $1-\frac{P_{receive}}{P_{send}}$.

### 6.3. Simulation Results

Average delay, total throughput and packet drop rate of three routing strategies are shown in Figure 4, Figure 5 and Figure 6. In general, interval Type-2 fuzzy set routing algorithm is more efficient than centralized DSP routing algorithm and ELB algorithm. Detailed analysis of the simulation results is shown below.

#### 6.3.1. Average Delay

Figure 4 shows the comparison of average delay for different sending rates.

As is shown in Figure 4, with the increase of individual data transmission rate, the average delay will increase. Under the small individual data transmission rate and light network load, the average delay of centralized DSP routing algorithm is larger than that of the other two algorithms. This is because interval type-2 fuzzy set routing algorithm and ELB algorithm can send the traffic via alternate paths, which greatly reduces the queuing delay. As individual data transmission rate increases, the load increases. Because the centralized DSP algorithm continues to try the same transmission path to reduce the average delay at the cost of the higher packet loss rate, the average delay of the centralized DSP algorithm is smaller than that of interval type-2 fuzzy set routing algorithm and ELB algorithm when the load is larger.

#### 6.3.2. Total Throughput

Figure 5 shows the comparison of total throughput for different sending rates.

As is shown in Figure 5, with the increase of individual data transmission rate, the total throughput will increase. Comparing the three routing algorithms, the total throughput of interval type-2 fuzzy set routing algorithm is the highest, followed by ELB algorithm and the total throughput of centralized DSP routing algorithm is the least. Under the same individual data transmission rate, the smaller the packet drop rate, the higher the total throughput in the same time. With the increase of individual data transmission rate, the traffic load increases accordingly. Because the centralized DSP algorithm continues to try the same transmission path and the packet drop rate also increases, the total throughput decreases accordingly. Compared with other routing algorithms, the total throughput of centralized DSP algorithm is the lowest.

#### 6.3.3. Packet Drop Rate

Figure 6 shows the comparison of packet drop rate for different sending rates.

As is shown in Figure 6, with the increase of individual data transmission rate, the packet drop rate will increase. Packet drop rate is caused by packet queues overflow. Comparing the three routing algorithms, the packet drop rate of interval type-2 fuzzy set routing algorithm is the lowest, followed by ELB algorithm and the packet drop rate of centralized DSP routing algorithm is the highest. With the increase of the load, the DSP always continues to try the same transmission path to transmit data, resulting in a large packet drop rate. The ELB algorithm adopts the corresponding congestion avoidance strategy according to the congestion state of adjacent satellites, while the interval type-2 fuzzy set routing algorithm can reduce congestion before congestion.

In conclusion, compared with traditional centralized DSP algorithm and ELB algorithm, the proposed interval type-2 fuzzy set routing algorithm can reduce the average delay, increase the throughput and reduce the packet drop rate.

## 7. Conclusions

In this paper, we propose an interval type-2 fuzzy set routing strategy to meet the multi-class QoS demand of SDSN. Firstly, the multi-QoS routing problem in SDSN is modeled. It is assumed that there are four neighbors of LEO satellite that does not pass through the polar region and there are four ISLs in total. There are four queues to store packets of four different types of applications in a LEO satellite, including time-delay sensitive application, video application, packet loss sensitive application and no special QoS requirements application. Then, the interval-type-2 fuzzy set routing algorithm is proposed to make fuzzy decisions. In the control plane, GEO satellites send control information to update routing table of LEO satellites. In the data plane, Interval Type-2 fuzzy set is used to make fuzzy decisions when selecting the next hop. When sender send a packet, the packet is placed in the corresponding queue of the sending buffer. The corresponding port is selected according to the routing table. The update of the routing table is based on Algorithm 1.

A series of experiments are conducted in Network Simulator (Version 2.35). As a result, by simulating Iridium system, compared with ELB routing algorithm and centralized DSP routing algorithm, the proposed in-terval type-2 fuzzy set routing algorithm can reduce average delay, increase total throughput and reduce packet drop rate.

## Figures and Tables

**Figure 1 sensors-21-06356-f001:**
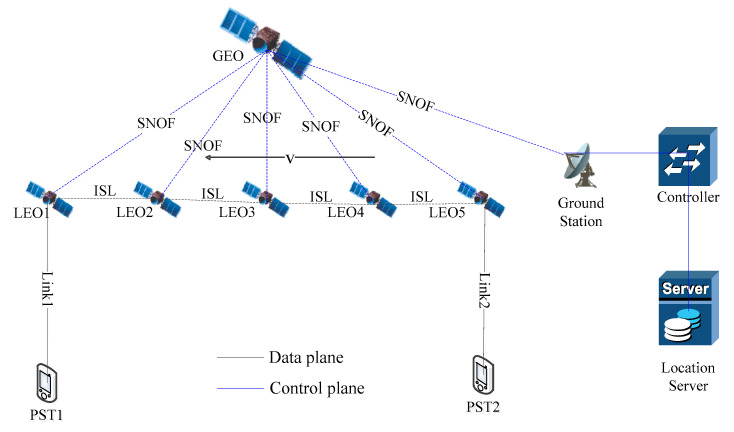
SDSN architecture.

**Figure 2 sensors-21-06356-f002:**
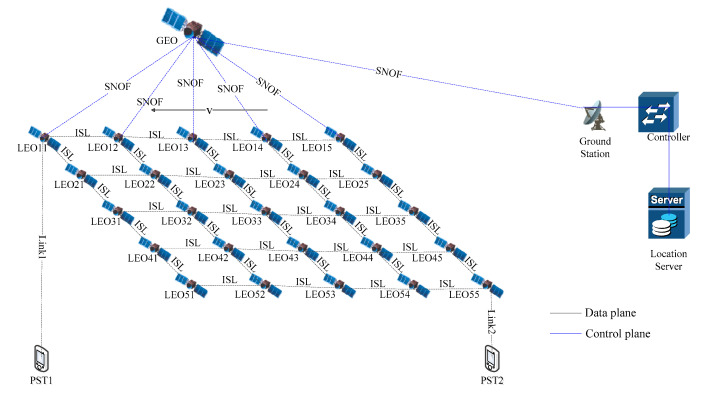
Multi-QoS routing model.

**Figure 3 sensors-21-06356-f003:**
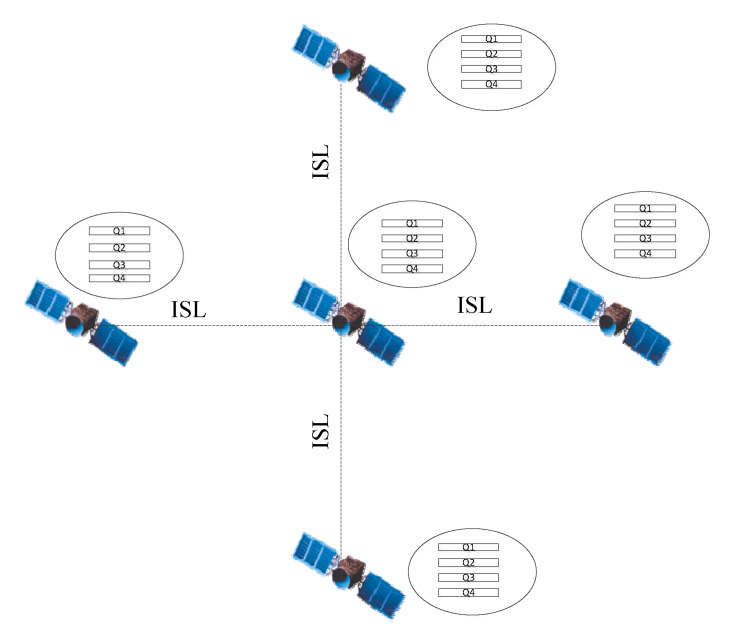
Queue model.

**Figure 4 sensors-21-06356-f004:**
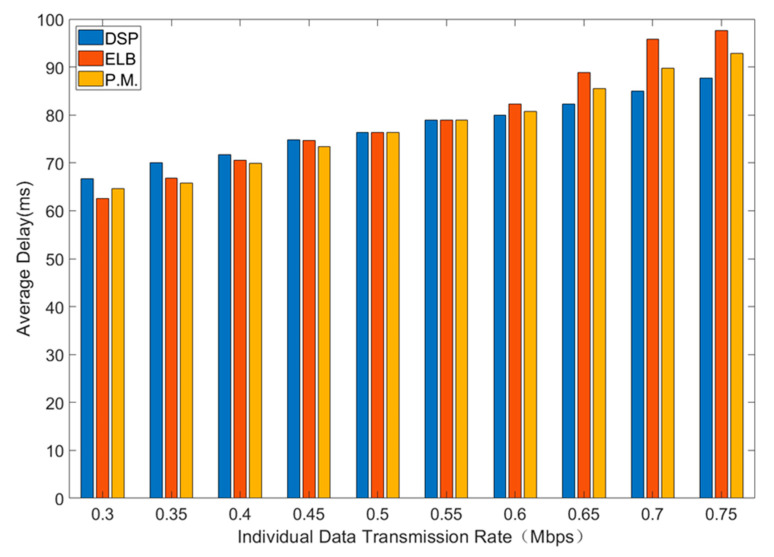
The average delay for different sending rates.

**Figure 5 sensors-21-06356-f005:**
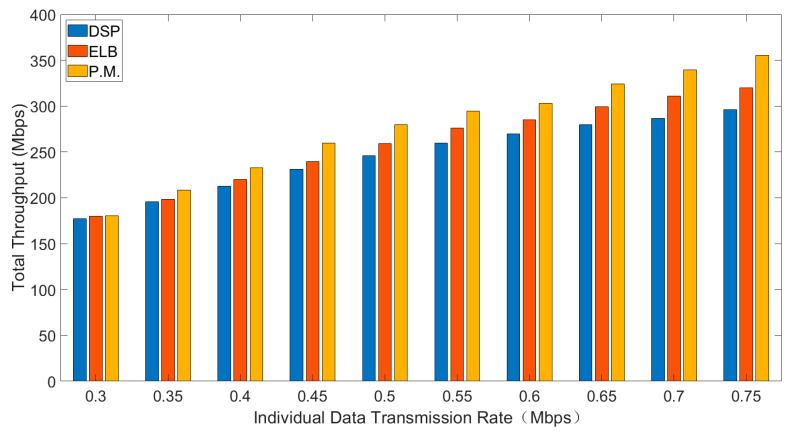
The total throughput for different sending rates.

**Figure 6 sensors-21-06356-f006:**
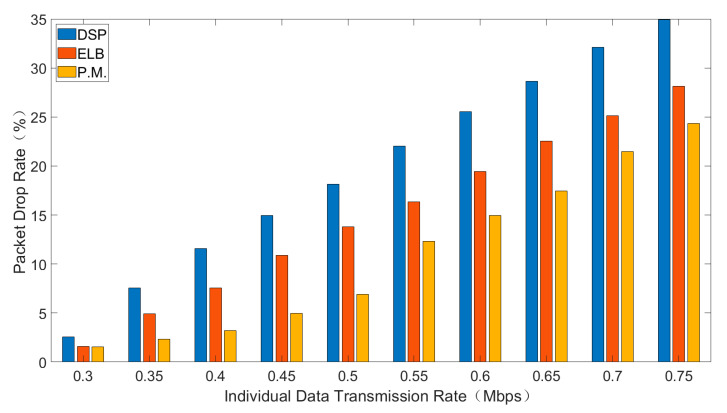
The packet drop rate for different sending rates.

**Table 1 sensors-21-06356-t001:** Linguistic term set and their corresponding values.

Label	Linguistic Terms	Corresponding IT2HFNs
AL	Absolutely low	(0.0, 0.0, 0.0, 0.0; 1,1) (0.0, 0.0, 0.0, 0.0; 1,1)
VL	Very Low	(0.0075, 0.0075, 0.015, 0.0525; 0.8,0.8), (0,0,0.02,0.07; 1.0,1.0)
ML	Slightly low	(0.2325, 0.255, 0.325, 0.3575; 0.8,0.8), (0.17,0.22,0.36,0.42; 1.0,1.0)
M	Middle	(0.4025, 0.4525, 0.5375, 0.5675; 0.8,0.8), (0.32,0.41,0.58,0.65; 1.0,1.0)
MH	Slightly high	(0.65, 0.6725, 0.7575, 0.79; 0.8,0.8), (0.58,0.63,0.80,0.86; 1.0,1.0)
H	High	(0.7825, 0.815, 0.885, 0.9075; 0.8,0.8), (0.72,0.78,0.92,0.97; 1.0,1.0)
VH	Very high	(0.9475, 0.985, 0.9925, 0.9925; 0.8,0.8), (0.93,0.98,1.0,1.0; 1.0,1.0)
AH	Absolutely high	(1.0, 1.0, 1.0, 1.0; 1.0,1.0), (1.0, 1.0, 1.0, 1.0; 1.0,1.0)

**Table 2 sensors-21-06356-t002:** The membership functions.

Measures	Membership Functions for Each Term of Each Measures							
*L_A_*	*g^8^(L_A_)* = AL	*g^7^(L_A_)* = VL	*g^6^(L_A_)* = ML	*g^5^(L_A_)* = M	*g^4^(L_A_)* = MH	*g^3^(L_A_)* = H	*g^2^(L_A_)* = VH	*g^1^(L_A_)* = AH
*L^B^*	*g^8^(L_B_)* = AL	*g^7^(L_B_)* = VL	*g^6^(L_B_)* = ML	*g^5^(L_B_)* = M	*g^4^(L_B_)* = MH	*g^3^(L_B_)* = H	*g^2^(L_B_)* = VH	*g^1^(L_B_)* = AH
*L^C^*	*g^8^(L_C_)* = AL	*g^7^(L_C_)* = VL	*g^6^(L_C_)* = ML	*g^5^(L_C_)* = M	*g^4^(L_C_)* = MH	*g^3^(L_C_)* = H	*g^2^(L_C_)* = VH	*g^1^(L_C_)* = AH
*L^D^*	*g^8^(L_D_)* = AL	*g^7^(L_D_)* = VL	*g^6^(L_D_)* = ML	*g^5^(L_D_)* = M	*g^4^(L_D_)* = MH	*g^3^(L_D_)* = H	*g^2^(L_D_)* = VH	*g^1^(L_D_)* = AH

**Table 3 sensors-21-06356-t003:** The values of parameters.

Parameter	$\bar{\lambda_{1}}$	$\underline{\lambda_{2}}$	$\bar{\lambda_{2}}$	$\underline{\lambda_{3}}$	$\bar{\lambda_{3}}$	$\underline{\lambda_{4}}$	$\bar{\lambda_{4}}$	$\underline{\lambda_{5}}$	$\bar{\lambda_{5}}$	$\underline{\lambda_{6}}$	$\bar{\lambda_{6}}$	$\underline{\lambda_{7}}$	$\bar{\lambda_{7}}$	$\underline{\lambda_{8}}$
Value	$\frac{1}{16}L$	$\frac{2}{16}L$	$\frac{3}{16}L$	$\frac{4}{16}L$	$\frac{5}{16}L$	$\frac{6}{16}L$	$\frac{7}{16}L$	$\frac{8}{16}L$	$\frac{9}{16}L$	$\frac{10}{16}L$	$\frac{11}{16}L$	$\frac{12}{16}L$	$\frac{13}{16}L$	$\frac{14}{16}L$

where *L* is the storage space length of the queue.

**Table 4 sensors-21-06356-t004:** The detailed parameters of Iridium [14].

	Iridium
Altitude	780 km
Planes	6
Satellites per plane	11
Inclination (deg)	86.4
Interplane separation (deg)	31.6
Seam separation (deg)	22
Intraplane phasing	Yes
Interplane phasing	Yes
ISLs per satellite	4
ISL bandwidth	25 Mb/s
Link bandwidth	1.5 Mb/s
Cross-seam ISLs	No
ISL latitude threshold (deg)	60

**Table 5 sensors-21-06356-t005:** Distribution of Traffic Flows (%).

**Source**	**Destination**					
	**N. America**	**S. America**	**Europe**	**Africa**	**Asia**	**Oceania**
N. America	60	10	15	2	10	3
S. America	35	40	12	2	8	3
Europe	40	5	40	2	10	3
Africa	40	2	30	20	5	3
Asia	20	2	10	2	50	6
Oceania	40	2	10	2	12	34

## Data Availability

No applicable.

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
