# Peer review of "Multi-Objective Optimisation in Multi-QoS Routing Strategy for Software-Defined Satellite Network"

_sensors, 2021, doi:10.3390/s21196356_

Round 1
Reviewer 1 Report
Minor comments:
- I will recommend you not to define and use abbreviatures in the abstract. Define and use them the first (and only one time) time you use them in the regular text. Once defined an abbreviature, please use it, i.e. Software-Defined Satellite Network. Define always abbreviatures in the same way, i.e. HDRP (Hierarchical and Distributed QoS Routing Protocol) is erroneus definition and different than other.
- Change works for contributions in the sentence "The main works are as follows"
- Avoid sentences like "There are also many QoS routing protocols and traffic balancing routing protocols." that are completly empty of content.
- Fix "voiceover internet protocol (VOIP)."
- Remove lines 147-149.
- Remove "The text continues here."
- I will suggest you to remove abbreviatures from the Title of sections.
- The title of Section 3 (Preliminaries) is not a good idea. Please change it.
- It is not neccesary to repeat severla times the same idea, like "[18] has proposed an architecture named software-defined satellite network (SDSN)."
- Figure 1 is to small to be viewed. Please fix.What doaes means intermittent lines in the Figure? Dta and control lines are not well defined in the Figure. Please fix.What is the function of SNOF in the Figure?
- Correct expressions like "Under the background of the above architecture".
Major comments:
- Introduction Section must identify problems and issues of Sattelite-SDN and then you must provide clearly the contributions of your paper that improve other solutions or represent an advance in research. Yo only have described what you presented in the paper (works). Please fix that.
- Consider the correctness of sentences like "The differ-ence between satellite network and terrestrial network is the dynamic network topology." due to there are terrestrial wireless networks that also are dynamic by nature.
- Review carefully the references because you used several ones incorrectly, i.e. [1] is used to reference Tora Algorithm and Industrial...
- The beginning of Section 3.1 is understandable because you referen sets that are not defined previouly... Please fix.
- To whih diagram did you refer when wrote: "The following diagram shows the routing model?
Reject comments:
- Please consider to change Section 3 in order it to be undestandable. In its current state it is un comprehensible.
- Please consider to change Section 5 becasue it is completdly undestandable. Moreover, Algorith 1 is not an algorithm. Please fix.
- Section 6 must be reformulated in order to consider a match with previous sections and comparison with other CONCRETE ROUTING ALGORITHM in Sattelite-SDN. Moreover, realistic condictions must be considered.
Reviewer 2 Report
see attached file

Reviewer 3 Report
The authors proposed an interval-type-2 fuzzy set routing algorithm to solve the problem of multi QoS routing in software-defined satellite networks.
The paper contains a lot of mistakes and strange terms like "dynamic topology" (meaning dynamic physical topology?), "fuzzy theory" (fuzzy sets theory?), and so on.
The paper needs intense "cleanup" and clarification in descriptions. Also, the conclusions should be more elaborated.
dynamic network topology ????
Wrong name:
Reviewer 4 Report
In this paper, the authors an multi-QoS routing scheme in the architecture of SDSN. Some suggestions are presented as follow.
- The authors are suggested to present current published routing schemes to highlight the main difference between the published routing schemes and the proposed scheme.
- The authors are suggested to identify some connections between fuzzy theory and the proposed routing scheme in Section 3. Only present fuzzy theory cannot support the idea behind the proposed scheme.
- All symbols and notation should be identified their physical meaning.
Round 2
Reviewer 2 Report
see attached file

Reviewer 3 Report
Authors fixed all mistakes, paper is ready to publish